# The Behavioural Impact on Cats during a Transition from a Clay-Based Litter to a Plant-Based Litter

**DOI:** 10.3390/ani12080946

**Published:** 2022-04-07

**Authors:** Jennifer Frayne, Michelle Edwards, James R. Templeman, Candace C. Croney, Sarah MacDonald-Murray, Elizabeth Flickinger, Adronie Verbrugghe, Anna K. Shoveller

**Affiliations:** 1Department of Animal Biosciences, University of Guelph, 50 Stone Road East, Guelph, ON N1G 2W1, Canada; jfrayne@uoguelph.ca (J.F.); jtemplem@uoguelph.ca (J.R.T.); smacdo20@uoguelph.ca (S.M.-M.); 2Ontario Agriculture College, University of Guelph, 50 Stone Road East, Guelph, ON N1G 2W1, Canada; edwardsm@uoguelph.ca; 3Primal Pet Foods, Primal Pet Group, 535 Watt Dr B, Fairfield, CA 94534, USA; 4Department of Comparative Pathobiology, Department of Animal Sciences, Center for Animal Welfare Science, Purdue University, 625 Harrison Street, West Lafayette, IN 47907, USA; ccroney@purdue.edu; 5Kent Pet Group, 2905 N Hwy 61, Muscatine, IA 52761, USA; elizabeth.flickinger@kentww.com; 6Department of Clinical Studies, University of Guelph, 50 Stone Road East, Guelph, ON N1G 2W1, Canada; averbrug@uoguelph.ca

**Keywords:** behaviour, cats, litter, transition

## Abstract

**Simple Summary:**

Environmental changes in the home, specifically around litter box management, can be stressful for cats, and resulting behavioural changes, such as house soiling, are one of the leading causes of owner frustration. Current guidelines recommend a 6-day litter transition; however, these recommendations are based largely on anecdotal reports. Our objectives were therefore to determine whether any behavioural changes occurred during a litter transition from clay-based to plant-based litter when following current transition guidelines and to identify behaviours that might signify a successful transition. Results presented in this study suggest that transitioning an adult cat from one litter product to another over 6 days is sufficient for maintaining normal litter box behaviours, but exposure to the new litter prior to replacement of the old litter should be recognized as a potential transition aid and warrants further investigation. Cats generally demonstrated increased interest and exhibited investigative behaviour (e.g., sniffing) towards the new litter during transition without showing behaviours that would indicate fear or aversion to the new litter product. The authors do want to recognize, though, that individual cat behaviour and potential stressors in the home environment must be taken into account when considering how to approach a litter substrate change.

**Abstract:**

Current guidelines recommend transitioning cats from one litter product to another over 6 days to minimize stress. The study objective was therefore to test these guidelines using 16 adult domestic cats (2 cohorts of 8) by observing behavioural changes associated with elimination throughout the litter transition. Cats were transitioned from a clay-based litter (CLAY) to a plant-based litter product (PLANT) over 6 days (period 1) via an incremental replacement of CLAY with PLANT. All cats then remained on PLANT for 8 days (period 2). This same transition process was executed for both cohorts, and litter box behaviours were observed via remote recording. Urination, defecation, cover, and dig behaviours were not different between periods 1 and 2 (*p* > 0.05). Sniffing frequency was greater in period 2 than period 1 (*p* < 0.05); however, during the litter transition (period 1), cats sniffed the litter boxes being transitioned from CLAY to PLANT more often and for longer than they did for the boxes consisting of only PLANT (*p* < 0.05). These data suggest that 6 days may be an adequate amount of time to transition a cat to a new litter, although successful transition may also be specific to the types of litters investigated.

## 1. Introduction

Cat owners and caregivers often believe that inappropriate behaviours appear suddenly in their cats. However, the distress that underpins these changes may exist for a long period of time before the owners realize there is a problem [1,2,3,4]. Behavioural changes in cats, such as house soiling, are one of the leading causes of owner frustration and often result in owners surrendering their cats to a humane society or animal shelter [2,3,4,5,6]. In fact, it has been reported that nearly 30% of cats that were surrendered had one or more behavioural concerns, and up to 43% of those were eliminating outside of the litter box [2]. Rehoming a cat with known issues of house soiling can be difficult and may result in euthanasia rather than re-homing [7,8].

Cats soil outside of the litter box for different reasons, but stress caused by conflict with other cats in the home or sudden environmental changes have been suggested to be the most prominent factors [3,9,10]. Additionally, litter box management is believed to play a considerable role in litter box aversion. Environmental changes in the home, such as owners being away from home for extended periods of time, greater excitement in the home or even sudden changes in litter substrate can increase stress in cats, resulting in aversion to the litter box [11,12]. As cats can have individual preferences, and home environments can vary, owners may find themselves testing out different litter products to find one their cat prefers and will use. Given the wide variety of features available in litter products, including clumping vs. non-clumping, scented vs. unscented, and clay vs. paper or plant-based substrates, there exists the potential for multiple litter transitions. To assist cats coping with a change in litter product, the American Association of Feline Practitioners (AAFP) and International Society of Feline Medicine (ISFM) recommend a transition period of 6 days from the old cat litter product to the new cat litter product [13,14,15,16]. However, this recommendation is based on anecdotal reports examining cat behaviour to determine the length of litter transition needed, and there remains a dearth of data-driven support for these recommendations [3,15,16,17,18].

Male cats have been identified as having a greater risk of house soiling, as well as having a reduced success rate of behaviour modification [19]. When studying specific cat behaviours, such as sniffing the litter box, it has been reported that male cats sniffed their environments more and exhibited different behaviours than did female cats [10], suggesting that male and female cats should be evaluated separately when comparing behaviours related to litter box use. Studies focusing on reduction of stress in the home are needed to further understand the complex nature of the domestic cat and to support good management practices to feline health and welfare [20,21].

Currently, clay cat litter is very popular on the market as this product is simple to use and cost-efficient for cat owners. Recently, though, largely due to public interest in environmentally friendly products, plant-based litter has become more popular and available for cat owners [22]. This litter type is comprised of sustainable sources, can be composted, and decomposes more rapidly than clay-based litter substrates [22]. Furthermore, if the litter is ingested by the cat or family dog, the product is better digested, reducing the risk of intestinal blockage.

The objectives of the current study were to determine whether any behaviour patterns changed during litter transition from clay-based (CLAY) to plant-based litter (PLANT) while following the current guidelines of a 6-day transition and to identify behaviours related to positive and negative affective states that would signal successful vs. unsuccessful transitions. One assumption we made regarding behavioural changes was that as the litter was transitioned, cats would have increased interest [10] but also greater stress [23,24]. Thus, we expected to see reduced dig and cover behaviours in the beginning of the transition period. As cats prefer to dig prior to using the litter box and then cover their eliminations after, failure to perform these behaviours has been associated with cats being dissatisfied with their litter box environments [10,25]. Dig and cover behaviours are presumably inhibited to escape the litter box area as quickly as possibly [10,25,26]. We hypothesized, therefore, that sniffing behaviours would be greater during the 6-day period of transitioning the cats from CLAY to PLANT (period 1) compared to the period following transition (period 2). Additionally, we hypothesized that the prevalence of cover and dig behaviours would be reduced due to increased stress during the early transitional stages of period 1. Last, we hypothesized that male cats would show more sniffing behaviour than female cats during the transition period, particularly during the early transitional stages.

## 2. Materials and Methods

All facilities and study procedures were approved by the University of Guelph Animal Care Committee (AUP#3972).

### 2.1. Study Subjects

The study included 16 cats with a mean age of 2.13 ± 1.41 years. Cats sourced from 2 local animal shelters and were divided into 2 cohorts of 8 cats based on shelter availability. The cats in cohort 1 included 3 neutered males and 5 spayed females (mean age of 2.63 ± 1.68) while cohort 2 included 4 neutered males, 3 spayed females, and 1 intact female (mean age of 1.63 ± 0.92). For more information regarding the cat demographics for each cohort, such as age, sex, and breed of each cat upon arrival, refer to Frayne et al. [27]. A licensed veterinarian examined all cats upon their arrival to ensure all were of good general physical health; however, no blood work or urinalysis was done. The cats participated in voluntary social interactions such as brushing, petting, and playing with the same caretakers two to three times a day during room and litter box cleaning, as well as during as an afternoon session on weekdays, that did not exceed 2 h of human presence in the cat room. Cats were acclimated to the room and to being group housed for 4 weeks prior to the study period beginning. For additional details regarding housing and enrichment, refer to Frayne et al. [27].

### 2.2. Litter and Litter Boxes

Eight uncovered litter boxes were arranged in a circle (Figure 1). The litter boxes in both cohorts were made from plastic with a high polish finish. For details regarding the litter box dimensions, refer to Frayne et al. [27]. For each of the 2 cohorts, at baseline, cats had been acclimated for 4 weeks to this litter box arrangement, with half of the litter boxes assigned to PLANT and half to CLAY. The CLAY treatment was a commercially available leading brand of unscented, clumping clay cat litter (Purina Tidy Cats Clumping Cat Litter; Nestlé Purina Petcare, Mississauga, ON, Canada). The PLANT treatment was a commercially available unscented, corn-based clumping cat litter (World’s Best Original Unscented Cat Litter; World’s Best Cat Litter, Muscatine, IA, USA). Both litter products were granular, with the PLANT treatment having a slightly larger particle size than the CLAY treatment. In addition, the PLANT treatment was approximately 50% lighter in bulk density (loose bulk density of 30.2 lb/ft^3^) than the CLAY. Twice per day, the litter boxes were scooped and all eliminations were recorded. All litter boxes included a 2-inch-deep layer of litter throughout the entire study period. All study subjects had been previously exposed to clay litter substrates from shelter they were sourced from; however, the authors do not know whether any of the cats had had previous experience with plant-based substrates or had pre-existing elimination-related behavioral issues. We acknowledge this lack of information as a limitation of the study.

The litter boxes were numbered 1–8, with boxes 1, 3, 5 and 7 containing CLAY litter and boxes 2, 4, 6 and 8 containing the PLANT litter (Figure 1) at the initiation of the study.

Baseline behaviours were recorded on day −1 with the above-described litter box arrangement. Period 1 consisted of the 6-day litter transition (days 0–5). During this period, CLAY litter boxes (litter boxes 1, 3, 5, 7) were transitioned over to PLANT litter. The PLANT boxes (litter boxes 2, 4, 6, 8) remained unchanged. Calculations were based on the assumption that the two litter types were thoroughly mixed prior to placing back in the circle. This mixture calculation was taken into account when adding the PLANT litter to ensure the litter box was of the right percentage required. The transition period started at baseline (day 0) with the removal of 25% of the CLAY litter and the addition of the equivalent volume of PLANT, then mixing the two together (Table 1). No change was made to the litter mixture (75% PLANT, 25% CLAY) on day 1, and days 0 and 1 were recorded as transitional stage 1. On day 2, mixed litter was removed and PLANT substrate was added to result in 50% of litter box containing PLANT litter and 50% containing CLAY. No change was made to the litter mixture (50% PLANT, 50% CLAY) on day 3, and days 2 and 3 were recorded as transitional stage 2. The addition of the PLANT substrate and removal of the mixture litter was repeated again on day 4 so as to confirm that the transitional boxes contained 75% PLANT and 25% CLAY. No change was made to the litter mixture (75% PLANT, 25% CLAY) on day 5, and days 4 and 5 were recorded as transitional stage 3. On day 6, boxes 1, 3, 5, 7 were completely emptied of the mixed substrate and filled with the equivalent amount of PLANT litter that was in the control boxes (boxes 2, 4, 6, 8). No change was made to the litter mixture (100% PLANT, 0% CLAY) on day 7, and days 6 and 7 were recorded as transitional stage 4. This ensured that each ‘transition stage’ consisted of 2 days. From day 7 onward, all 8 litter boxes contained only the PLANT litter.

### 2.3. Behavioural Data Collection

For detailed information regarding the video-recorded behavioral data collection, refer to Frayne et al. [27]. The 2 coders utilized were considered to have excellent reliability characterizations (beta-score of greater than 0.75) and could code each cohort independently [28].

### 2.4. Behaviour Assessment and Coding

The behaviour ethogram utilized for this study was modified from McGowan et al. [10] to include 10 primary behaviours based on litter box involvement only. For an overview of the modified ethogram utilized, refer to Frayne et al. [27]. In addition to recording each behaviour occurrence, the time, duration, and location of the litter box that was used was also noted. To be considered an event, there must have been a minimum of one behaviour observed that was listed in the ethogram. An event was considered concluded when the cat left the litter box area.

### 2.5. Statistical Analyses

Data were analyzed using a mixed model which included the effect of the week, litter box location, and their interaction as fixed effects, with cats within cohorts as a random variable. Sniff frequency and duration data were evaluated utilizing a Log-normal distribution while defecation and urination frequency and duration data were modeled using a Gaussian distribution based on meeting the assumptions of the model. Analyses of variance were conducted, and least square means presented in Table 2 and Table 3. The total duration was defined as the total amount of time spent performing the respective behaviour for each treatment used over the entire study period analyzed (14 days). Duration was defined as the length of time (in seconds) a cat was observed performing a behaviour. The mixed model was analyzed using a GLIMMIX procedure in SAS (v. 9.4, SAS Institute, Carey, NC, USA), while a correlation analysis was used to evaluate relationships among the behaviours was conducted using PROC CORR in SAS. Results were statistically significant at *p* < 0.05.

## 3. Results

No correlations were observed for any behaviour analyzed (urination, defecation, cover, dig, sniff) among treatment periods (*p* > 0.05). These data suggest that behaviour did not change as CLAY boxes were transitioned to PLANT.

### 3.1. Urination and Defecation Behaviour

There were no differences observed in the number of urinations or defecations per day or duration of urination or defecation behaviours between periods, transitional stages within the period, cohorts, or sexes (*p* > 0.05).

### 3.2. Sniffing Behaviour

When sniff-post and sniff-pre behaviours were analyzed separately, no differences were reported (*p* > 0.05). As such, using a Log-normal distribution for total sniff duration and Gaussian distribution for sniff frequency, all sniff behaviours were analyzed together. When analyzing frequency behaviours of male and female cats together, there were no differences due to transition stage within cohort, or litter box location (*p* > 0.05). The cats were found to sniff the litter boxes in period 2 significantly more than in period 1 (period 2, 6.79 + 0.79 times; period 1, 5.59 + 0.80 times; *p* = 0.02). However, during the litter transition (period 1), the cats were found to sniff the litter boxes being transitioned from CLAY to PLANT more frequently (*p* < 0.01; Table 3) and for a longer duration (*p* < 0.01; Table 3) than the original (unchanged) PLANT litter boxes. No differences were observed for the total duration of sniff behaviours by transition stage or day within transition stage (*p* > 0.05). For total duration of sniff behaviour of male and female cats together, there were no differences reported between periods or between the transition stage, day within transition stage, or litter box location (*p* > 0.05).

### 3.3. Dig and Cover Behaviour

No differences were observed for either frequency or duration of dig or cover behaviours between periods, transitional stages within the period, cohorts, or sexes (*p* > 0.05). It should be noted, though, that 1 cat in cohort 2 (female, 3 years of age, domestic short hair) did not exhibit any cover or dig behaviour during the entire study period.

### 3.4. Eliminations Outside of the Box

During the entire 14-day (transition and post 8 days post-transition) study period, a total of four urinary eliminations and nine fecal eliminations outside the box were recorded. For cohort 1, four urinary eliminations and eight fecal eliminations were found in the room outside of the litter box. For cohort 2, only one fecal elimination was found outside of the litter box. One cat (female, 1 year of age, domestic short hair, spayed) in cohort 1 was observed on video urinating and defecating outside of the litter boxes on camera on days 7, 8, and 9, when all boxes contained 100% PLANT litter. Cats were recorded within the litter box area during recording hours; however, the rest of the room was not recorded during the study and therefore individual cats responsible for eliminations elsewhere in the room could not be identified.

## 4. Discussion

The results of the present study support the notion that when a new litter substrate is offered for investigation prior to removal of an old litter and the old litter is removed over the recommended duration of time (6 days) that cats do not significantly change their behaviour related to litter box use, aside from sniffing behaviour. Current recommendations by litter companies as well as the AAFP and ISFM suggest transitioning cats from one litter to another litter product over 6 days to minimize the stress on the cat and limit neophobic behaviour [3,15,16,17,18]. It may be also recommended to allow investigation of new litters prior to litter change, but this requires further investigation.

Prior experience of the cats with clay litter may have influenced their preferences when comparing CLAY litter to PLANT [18], which is primarily why this was not specifically examined in this study. As the cats were originally offered both PLANT and CLAY during acclimation to the environment, it is unlikely that they did interact with the PLANT litter as a result of novelty. This design was purposely chosen to ensure that we did not merely measure novelty, and so, the current study truly measured the removal of clay-based litter. Overall, the cats sniffed the litter boxes being transitioned from CLAY to PLANT more times than the original PLANT litter. Sniffing behaviour in cats is often overlooked when evaluating a particular cat’s response to an environment and can lead to underestimating how important the olfactory environment is cats [6]. The olfactory epithelium of cats is nearly 7 times larger than that of a human, and as such, cats are more sensitive to olfactory changes than their human caretakers [29]. Very little research has been published on how a cat interprets a new odor, but they have been reported to be sensitive to novel smells [30,31]. If cats find a smell aversive, they will often avoid the area entirely [10,30].

As male/female cat dyads have been found to sniff more in general than male/male or female/female pairs [32], this could explain why our sniffing behaviours were more prevalent than those observed in previous studies, since both cohorts were mixed sexes in the present study. The cats were also found to sniff the litter boxes more during period 2 when all boxes were filled entirely with PLANT litter. In period 1, the cats sniffed the litter boxes being transitioned from CLAY to PLANT more than the control (PLANT only) litter boxes. As such, it was expected that the cats would exhibit greater overall sniff behaviour in period 1. Since the cats still showed interest in the litter following the period of transition, they appeared to adapt to the change in litter products and did not show increased aversive behaviours towards PLANT. As our cats were acclimated to the novel litter (PLANT) prior to the transition period, our findings suggest that allowing cats in the home to become accustomed to the new litter product prior to transitioning from their previous litter type may improve their response to the transition. This would be a novel addition to recommendations for a litter product transition and would indicate that an additional box with the new litter should be provided alongside the old litter, followed by gradual addition of the new litter to the old litter boxes until the transition is complete.

In the present study, transitioning the cats from a CLAY litter to a PLANT litter resulted in no change to behaviours that would suggest stress associated with urinary and fecal elimination. The finding of no difference in cover and dig behaviours throughout the study may suggest that the cats in the present study were minimally or not stressed during the transition period and that this resulted in no change in their cover and dig behaviours. Situations with additional stressors, such as sudden or drastic changes in a regular home environment, may impact the length of transition time needed to change litter substrate. When determining the length of litter transition, the current behaviour of the cat, changes in the home, and the cat’s health all need to be considered.

No differences in behaviour were observed between sexes in the current study, an outcome that was in agreement with data reported in other similar studies [33,34]; however, a greater number of cats may be required to fully validate this outcome. There is a dearth of available literature investigating litter box use in cats, especially in a laboratory setting. Therefore, further research is needed to determine whether male and female cat behaviours differ and, if so, what environmental factors may alter these responses.

Eliminations outside of the litter box were minimal in this study, but as the cats’ history of litter box use was unknown, this group of subjects could have been more resilient to litter product changes [21,23]. Gradual changes allow cats to adapt to their surroundings [3,18], and because of minimal urination outside the litterbox, the 6-day transition period in the present study was deemed appropriate [3,9,18]. However, nervous or easily stressed cats may require a longer time to transition than more resilient cats. The cat that was videotaped defecating out of the litter box (female, 1 year of age, domestic short hair, spayed) may have been more distressed than those who used the litter boxes consistently [9,10]. Prior assessment of this cat’s litter box usage and personality before transitioning would have provided more of a baseline to compare the behaviours during the transition period, but this was not the focus of the current study. Fecal elimination outside of the litter box is reported less often than house soiling by owners [3], suggesting that a cat that does exhibit this behaviour may be highly distressed. One cat from cohort 2 (female spayed, 3 years of age, domestic short hair) did not exhibit any cover or dig behaviour during the entire study duration, regardless of stage of litter transition, which suggests that this individual cat may have been under more stress than in the other cats [10]. The circular litter box arrangement could have been a potential stressor for the individual cats as well, as the most favoured position for cats to eliminate is against a wall, presumably to safely eliminate while observing their surroundings [35]. With only one litter box closest to the wall, this may have been an issue for a more environmentally sensitive cat. In fact, similar to the other cats, the cat that was identified as not exhibiting any cover or dig behaviours preferred the spot against the wall (litter box 3; Figure 1) most often. Moreover, the location of the entire group of litter boxes may have been another potential stressor, as they were all placed in the same spot and were all in close proximity to each other. This litter box orientation may have resulted in more environmentally sensitive cats having to leave a space they perceive as safe in order to eliminate in one of the designated litter boxes, which could have further contributed to the apparent preference for the box closest to the wall.

In the present study, most cats appeared to acclimate successfully to the litter transition, as demonstrated by lack of difference in behaviours associated with stress; however, data from these two cats support the notion that individual preference and personality need to be considered when changing a cat’s environment. Having the opportunity for multiple cats to use the litter boxes simultaneously is a potential stressor that would only be present in multi-cat homes. Since this is not representative of individually housed cats, it is acknowledged as a potential limitation of this study. Resource guarding by more dominant cats may result in more timid cats shortening their interaction to reduce the likelihood of an altercation [23].

Observation of individually housed cats rather than cats in a group situation is also warranted to identify any sex differences with regards to assessing the behavioural response to litter adaptation. Indeed, another limitation of this study is the relatively small sample size and the individual sampling of the cats within a group-housed situation. Group-housing of cats can be a stressor depending on the population of cats; however, in this study, stress levels were observed to be low in terms of behaviour changes when interacting with the litter box. Because the cats in the current investigation were acclimated to the environment and had sufficient resources to minimize negative affective states, it is unsurprising that little social conflict occurred. Very few aggressive altercations around the litter box were identified on the video recordings and cats did have the ability to enter the litter box from more than one area of the room, allowing them to avoid interactions. An additional limitation of this study is the lack data on the cats’ litter box behaviour history. As the cats were sourced from animal shelters, the behavioural history is uncertain as some were surrendered, and others arrived as strays. This lack of background information results in unanswered questions regarding prior socialization of the cats, their resilience to change, and their familiarity with different types of litters. Moreover, as only adult cats were used, inferences to how kittens would respond to litter change cannot be made from the data presented herein. Kittens typically are more interested in novel objects [12], so they may be more likely to try new litter products and have fewer established preferences due to having a shorter learning history with specific litter products. Finally, in this study we did not assess cat temperament prior to starting the transition and this may have given us additional variables to understand the response to litter box change. Identifying the temperament of cats as well as their individual coping styles might have permitted us to determine if their individual behaviours in the litter box correlated to their coping styles. Evaluating the coping styles of the cats might allow us to identify which are proactive or reactive in response to environmental change and could enhance our ability to identify indicators of stress in response to litter and other changes [5]. This could potentially assist owners in identifying cats that may exhibit more severe behaviour fluctuations when changes occur in the home.

## 5. Conclusions

The results presented herein suggest that transitioning an adult cat from one litter product to another over 6 days is sufficient for maintaining normal litter box behaviours, but exposure to the new litter prior to replacement of the old litter should be recognized as a potential aid in transition and warrants further investigation. Cats generally demonstrated increased interest and exhibited investigative behaviour towards the new litter during transition without showing an increase in neophobia or significant elimination behaviour changes. The authors also acknowledge that individual cat behaviour needs to be taken into account when considering how to approach a litter substrate change in a manner that is minimally disruptive to cats.

## Figures and Tables

**Figure 1 animals-12-00946-f001:**
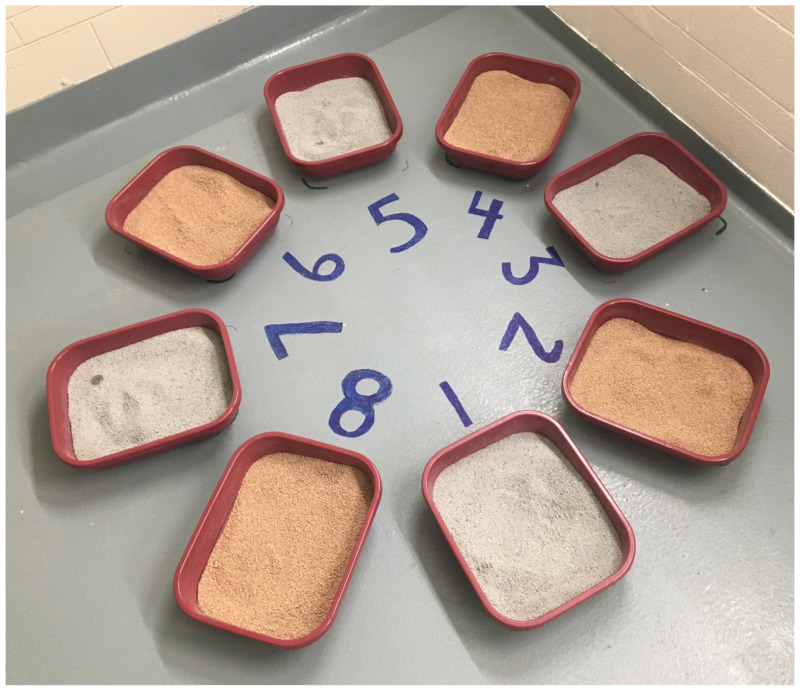
Arrangement of litter boxes containing alternating clay and plant-based litter.

**Table 1 animals-12-00946-t001:** Description of the study timeline including the litter box transition period (period 1) and the associated litter box transition calculations for litter boxes 1, 3, 5, and 7 based on volume.

Day	Period	Transition Stage	Mixture	Mixture Removed	Litter Added
−1 *	0 (pre transition)	0	(100% clay)	-	-
0	1 (transition)	1	(75% clay/25% plant)	Remove 6.5 cups ^+^ clay	Add 7 cups ^+^ plant
1	1 (transition)	1	(75% clay/25% plant)	No change	No change
2	1 (transition)	2	(50% clay/50% plant)	Remove 9.25 ^+^ cups	Add 9.75 cups ^+^ plant
3	1 (transition)	2	(50% clay/50% plant)	No change	No change
4	1 (transition)	3	(25% clay/75% plant)	Remove 13.5 cups ^+^	Add 14 cups ^+^ plant
5	1 (transition)	3	(25% clay/75% plant)	No change	No change
6	2 (post transition)	4	(100% plant)	Dump whole litter box	Add 28 cups ^+^ plant
7	2 (post transition)	4	(100% plant)	No change	No change
8	2 (post transition)	-	(100% plant)	-	-
9	2 (post transition)	-	(100% plant)	-	-
10	2 (post transition)	-	(100% plant)	-	-
11	2 (post transition)	-	(100% plant)	-	-
12	2 (post transition)	-	(100% plant)	-	-
13	2 (post transition)	-	(100% plant)	-	-

* Day −1 was not included in the statistical analyses. ^+^ 1 cup = 250 mL or 8 fluid ounces.

**Table 2 animals-12-00946-t002:** Frequency of sniff events per litter box over the 6-day transition period.

Litter Box	Litter Type	Frequency of Sniff Events	Standard Error
1	Transition from clay to plant	7.6 ^a^	1.15
2	Plant	2.1 ^b^	1.36
3	Transition from clay to plant	8.5 ^a^	1.15
4	Plant	2.0 ^b^	1.31
5	Transition from clay to plant	8.9 ^a^	1.18
6	Plant	3.8 ^b^	1.15
7	Transition from clay to plant	8.9 ^a^	1.18
8	Plant	2.8 ^b^	1.48

^a,b^ Means with no alike superscript are different (*p* < 0.05).

**Table 3 animals-12-00946-t003:** Mean total duration of sniff events per litter box over the 6-day transition period.

Litter Box	Litter Type	Duration of Sniff Events (Seconds)	Standard Error
1	Transition from clay to plant	49.0 ^a^	12.98
2	Plant	9.5 ^b^	3.09
3	Transition from clay to plant	63.9 ^a^	16.93
4	Plant	7.8 ^b^	2.42
5	Transition from clay to plant	72.2 ^a^	19.62
6	Plant	13.8 ^b^	3.65
7	Transition from clay to plant	74.6 ^a^	20.26
8	Plant	9.0 ^b^	3.19

^a,b^ Means with no alike superscript are different (*p* < 0.05).

## Data Availability

The data presented in this study are available on request from the corresponding author.

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
