# Peer review of "The Behavioural Impact on Cats during a Transition from a Clay-Based Litter to a Plant-Based Litter"

_animals, 2022, doi:10.3390/ani12080946_

Round 1

Reviewer 1 Report

Overview and general recommendation:

This manuscript investigates the behaviour of cats during a litter transition from clay-based to plant-based litter when following current transition guidelines. Another objective of the paper is to identify cats’ behaviours that might signify a successful transition. Environmental changes in the home, such sudden changes in litter substrate can increase stress in cats, resulting in aversion to the litter box. Thereby, the paper covers a very interesting topic in the field of cat’s management and behaviour, which is suitable for this journal, and refers to most relevant literature in the area. The article is clearly laid out and all the elements are present. There are just a few issues with this manuscript that should be addressed.

Specific comments

Keywords

-I would delete “video observation”: is not really the focus of the paper; “video observation” is just a method used to evaluate the real objective of the study.

Materials and Methods

-Lines 123-125: You state “No bloodwork was taken; however, the 123 cats were monitored closely by the caretakers for any possible health complications prior 124 to the study starting”, but did you take a urine test?

It would have been interesting to rule out urinary tract problems that may also support inappropriate elimination behaviour.

-Lines 125-127: if I understood well the Feliway diffuser is plugged in 6 days, one before the arrival of the cats and for other 5 days. Then did you remove it for the rest of the study? It is correct?

-Lines 127-129: would it be possible to insert a picture of the room?

Results

-Lines 244-248: in this paragraph there is an interpretation of the results and should be moved to the discussion session

-Lines 286-294: didn’t you recorded any elimination outside the litterboxes in the second seven days of the study after the 7 days of transition? Have you recorded any elimination outside the litterboxes in the 4 weeks of acclimatation? Do you have any information regarding any previous inappropriate elimination of cats in the shelter?

Tables

Table 2: if I correctly understood, the litter box transition occurred in 7 days, and over the next seven days you just evaluated the behavior of cats. Is that correct? If yes, it would be better to delete these last seven days from the table or to modify the caption of the table.

Author Response

Please see the attached document with our point-by-point responses to all of your comments.

Reviewer 2 Report

The manuscript is relevant to the field of animal behavior science, in particular to clinical behavior, and is well written and structured. I have, however, a major question on the scope of the manuscript and the data presented and discussed. The Author says that the main goal of the manuscript is to investigate if the recommended transition period of 6 days from one litter to the other is adequate. With this purpose in mind, cats were exposed to a number of litter boxes used to transition the litter from CLAY to PLANT and to boxes containing only the new litter, PLANT, since the beginning of the study. From the very scarce data presented on the frequency with which the cats used the litter boxes, just 3 lines in section 3.1, I understand that the PLANT boxes presented from the beginning of the study are used as frequently as the transition litter boxes. If this is correct, I would say that one of the main findings in the study is that the transition may not be necessary at all (for the cats in this study at least). Why is this not covered at all in the manuscript? These findings should be better illustrated in the Results section and addressed in the Discussion and Conclusion.

If instead, I understand wrongly and there is a difference in the frequency of use between PLANT and the transition litters, these data must be presented more in details to make this point clear. There is an abundance of data on “sniffing” and some unnecessary data (i.e. Table 1 that may be substituted with a short sentence of the group composition), but no accurate data on the frequency of use.

Here a few more comments on the manuscript:

  • 93-95 – This hypothesis should be supported with an early explanation of the rationale behind it. The Author does this later in ll.337-342 of the discussion. I recommend moving this part of the discussion to the introduction.
  • 123 – I would mention here that no bloodwork and urinalysis was done. Because the topic of this paper is elimination, I think urinalysis is another relevant test that could have been performed.
  • 125 – This product, Feliway Friends, is marketed with a different name in other countries. I would add here who distributes the product and in what country. I guess CEVA, Canada? If this information is available, I would also list the composition of the product so readers can identify what the equivalent product is in their country.
  • 155-157 – The setup explained in “At baseline,…” was the same for the 2 cohorts, right? I would open this sentence with “For each of the 2 cohorts, at baseline….”.
  • 185-187 – I guess the calculation accounted for the fact that 50% of the litter left in the box was already a mixture of the two litterboxes. Right? I would explain this in the text. I would also specify in the corresponding table what the volume in a cup is, as I do not think this is an international standardized measure.
  • 289 – Is this the one intact female? Later in the discussion, the Author mentions another female cat (l.369). Are these both intact (non-spayed) females? I understand only one intact female was included.
  • 372-376 – Another potential stressor associated with the location of the litter boxes is that they were all placed in the same spot and very close to each other. Litter boxes must be placed in or close to the core area where a cat spends most of its time and where it feels safe. Fearful cats in the study had to leave their safe core area to eliminate, which may have put them in a vulnerable and unsafe condition (as perceived by the cats themselves). This may also explain why some cats preferred boxes close to the wall, as they provide a safer “hiding” place and they do not have to worry about other cats approaching from the side protected by the wall.

Author Response

(The authors gave the same response as above.)

Reviewer 3 Report

The authors convinced me in their introduction that this was an important topic from an animal welfare point of view.  The study design is appropriate, but the sample size is small.  This latter point is acknowledged in the discussion.

The study could have gained more value by pre-testing each cat for behavioural characteristics and for stress levels during the study.  This would have allowed more information to be gleaned from the study and allowed the authors to make more conclusions about the generality of their findings.

Overall I am satisfied the design and analysis of the study is appropriate and the authors have drawn valid conclusions based on their data and acknowledged the several limitations of the study design.

Author Response

(The authors gave the same response as above.)
